# Mandelbrot and Julia Sets of Transcendental Functions Using Picard–Thakur Iteration

**Ashish Bhoria** [1] , **Anju Panwar** [1,*] **and Mohammad Sajid** [2,*]

1    Department of Mathematics, Maharshi Dayanand University, Rohtak 124001, India;
     ashish.rs.maths@mdurohtak.ac.in
2    Department of Mechanical Engineering, College of Engineering, Qassim University,
     Buraydah 51452, Saudi Arabia
*    Correspondence: dranjupanwar.math@mdurohtak.ac.in (A.P.); msajd@qu.edu.sa (M.S.)

**Abstract:** The majority of fractals' dynamical behavior is determined by escape criteria, which utilize various iterative procedures. In the context of the Julia and Mandelbrot sets, the concept of "escape" is a fundamental principle used to determine whether a point in the complex plane belongs to the set or not. In this article, the fractals of higher importance, i.e., Julia sets and Mandelbrot sets, are visualized using the Picard–Thakur iterative procedure (as one of iterative methods) for the complex sine $T_c(z) = asin(z^r) + bz + c$ and complex exponential $T_c(z) = ae^{z^r} + bz + c$ functions. In order to obtain the fixed point of a complex-valued sine and exponential function, our concern is to use the fewest number of iterations possible. Using MATHEMATICA 13.0, some enticing and intriguing fractals are generated, and their behavior is then illustrated using graphical examples; this is achieved depending on the iteration parameters, the parameters '$a$' and '$b$', and the parameters involved in the series expansion of the sine and exponential functions.

**Keywords:** algorithms; fractals; Julia set; Mandelbrot set; Picard–Thakur iteration; escape criterion; iterative methods

**MSC:** 28A80; 37F10; 37C25

## 1. Introduction

Fractals are mathematical patterns that repeat themselves at different scales, creating intricate and often beautiful shapes. They are found in many natural phenomena, such as the branching of trees, the formation of snowflakes, and the patterns on a fern leaf. Fractals can be generated using iterative schemes (or iterative methods), where a simple equation or set of rules is repeated many times over to create complex and self-similar patterns. The most famous example of such a scheme is the Mandelbrot set [1], which is generated by iterating the function $z = z^2 + c$, where $c$ is a complex constant. Other fractals can be generated using different iterative methods. For example, the Julia set [2] is generated by iterating the equation $z = z^2 + c$, where $c \in \mathbb{C}$ and $z$ is initialized with a different value for each point on the complex plane. The resulting plot shows which points on the plane generate fractal patterns and which do not. In fractals, fixed point theory is a key idea, as it provides a way through which to understand the iterative schemes that generate fractals. In the context of fractals, the fixed points of an iterative scheme are the points in the fractal that remain unchanged no matter how many times the scheme is iterated. These fixed points are often called attractors as they "attract" nearby points in the fractal toward themselves. In order to color and visualize fractals, the idea of "escape" is crucial. Often, a color is assigned to a point that escapes to infinity quickly, but other colors are assigned to other points that do not escape or escape more slowly. The Mandelbrot set's rich and exquisite patterns can be seen in the images that the escape time algorithm produces. The escape time algorithm is affected by the maximum number of iterations required to determine whether or not

the orbit sequence approaches infinity. This approach provides a helpful mechanism that is used to illustrate some dynamic system characteristics when using an iterative process. Initially, the Mandelbrot and Julia sets were obtained using a straightforward Picard iterative technique for the polynomial $T_c(z) = z^2 + c, c \in \mathbb{C}$. By utilizing several iterative techniques as the Mann [3–5], Ishikawa [6], Noor [7], Picard–Ishikawa [8], Jungck–CR [9] iterations, certain fractals of a general character have also been constructed. Due to the fact that iterative procedures are known to result in variations in form, size, color, and other aspects for the same function, other researchers [10–14] have later employed various iterative techniques (or iterative methods) to generate variations of these sets, as well as sought to analyze their dynamics and pattern for various polynomials.

The Julia and Mandelbrot sets are typically associated with complex quadratic polynomial iteration. However, comparable approaches can be used to investigate the behavior of complex transcendental functions. Romera et al. [15] employed the Picard iterative technique to study complex families, whereas Prasad et al. [16] adopted the Ishikawa iterative scheme to obtain identical outcomes. Ref. [17] investigated the dynamics of a new generalized entire transcendental function using Mann iteration. Qi et al. [18] used different iterations to illustrate certain fixed point results for a sine function. In [19], complex polynomials involving sine and cosine functions via the Picard–Mann orbit were calculated. Ref. [20] used a four-step iteration with s-convexity to demonstrate the Julia and Mandelbrot sets of complex cosine functions. Tassaddiq et al. [21] used DK iteration to produce the Julia and Mandelbrot sets with complex sine $T_c(z) = sin(z^r) + c$ and exponential functions $T_c(z) = e^{z^r} + c$. Tanveer et al. [22] utilized the Mann and Picard–Mann iterations for the function $z^p + \log c^t$ to study the Mandelbrot set. In this research, the Picard–Thakur iterative scheme was used to generate fractals using complex sine $T_c(z) = asin(z^r) + bz + c$ and complex exponential functions $T_c(z) = ae^{z^r} + bz + c$. It is worth mentioning that the Julia and Mandelbrot sets associated with complex transcendental functions can display beautiful and intricate fractal patterns that are similar to the conventional Julia and Mandelbrot sets. Exploring these sets can provide beautiful visual representations of mathematical concepts, as well as insights on the behavior of complex transcendental functions.

As the function $T_c(z)$ contains a particular type of sine and exponential functions, all of the fractals generated in this study were exceedingly novel, appealing, and pleasant. While other iterative techniques (or iterative methods) like Mann [17], Picard–Mann [19], DK [21], etc., require more iterations, the Picard–Thakur orbit only needs 10 iterations for a decent approximation of fractals. In [23], the author introduced the concept of the embedding of memory in complex maps and fractal generation, which typically refers to the way in which previous iterations are stored and influence subsequent iterations. Embedding memory can also involve modeling the fractal generation process as a dynamical system.

The fractals generated in this work provide a visual representation of how complex sine and exponential functions behave under iteration, which can be valuable for understanding their dynamics. Fractals can also be used as a basis for creating digital art, antenna design [24], image compression [25], image encryption [26], and even music composition [27].

The rest of the article is structured as follows. In Section 2, several key definitions of the Mandelbrot and Julia sets, as well as the proposed iteration, are provided. In the generation of fractals for complex sine and exponential functions, Section 3 offers some fixed point results. In Section 4, using suggested algorithms, we discuss how the Julia and Mandelbrot sets behave with various parameters. In Section 5, we draw the conclusions of this article.

## 2. Preliminaries

**Definition 1** (The Picard orbit [28]). *Consider $\mathbb{Y}$ to be a non-empty set and $h : \mathbb{Y} \to \mathbb{Y}$. Picard's orbit for any initial point $z_0 \in \mathbb{Y}$ can be defined as the set of all iterates of $z_0$, i.e.,*

$$O(h, z_0) = \{z_n : z_n = h(z_{n-1}), n = 1, 2, 3, \dots \}.$$

**Definition 2** (The filled Julia set [28]). *For the function $T_c(z) = z^r + c$, the filled in Julia set is defined as*

$$K(T_c) = \{z \in \mathbb{C} : T_c^i(z) \text{ does not approach to } \infty\}$$

*where $T_c^i(z)$ is the i-th iterate of function $T_c$, and the Julia set for polynomial $T_c(z) = z^r + c$ is the boundary of $K(T_c)$, that is,*

$$J(T_c) = \partial K(T_c).$$

**Definition 3** (The Mandelbrot set [28]). *The Mandelbrot set M is made up of all the parameter values $c \in \mathbb{C}$ for which the filled in Julia set of $T_c$ is connected, i.e.,*

$$M = \{c \in \mathbb{C} : K(T_c(0)) \text{ is connected}\}.$$

*Equivalently,*

$$M = \{c \in \mathbb{C} : T_c^i(0) \nrightarrow \infty \text{ as } i \to \infty\}.$$

**Definition 4** (The Picard–Thakur orbit [29]). *Let $\mathbb{K}$ stand as a subset of complex numbers, and $T_c \colon \mathbb{K} \to \mathbb{K}$ is a mapping. For the initial point $z_0 \in \mathbb{K}$, the Picard–Thakur orbit is defined as*

$$
\begin{aligned}
z_{n+1} &= T_c(u_n), \\
u_n &= (1 - \alpha_1) \, T_c(w_n) + \alpha_1 T_c(v_n), \\
v_n &= (1 - \beta_1) \, w_n + \beta_1 T_c(w_n), \\
w_n &= (1 - \gamma_1) \, z_n + \gamma_1 T_c(z_n),
\end{aligned}
\tag{1}
$$

*where $\alpha_1, \beta_1,$ and $\gamma_1 \in (0,1)$. For simplicity, we take $z_0 = z$, $u_0 = u$, $v_0 = v$, and $w_0 = w$. The above sequence of iterates given by Equation (1) is known as the Picard–Thakur orbit, which can be written as $PTO(T_c, z_0, \alpha_1, \beta_1, \gamma_1)$.*

## 3. Main Results

In this section, we prove the general escape criterion for transcendental complex functions, $T_c(z) = a\sin(z^r) + bz + c$ and $T_c(z) = ae^{z^r} + bz + c$ where $r \geq 2, c \in \mathbb{C}$.

*3.1. Escape Criterion for $T_c(z) = a\sin(z^r) + bz + c$*

Since,

$$
\begin{aligned}
|\sin(z^r)| &= \left| \sum_{m=0}^{\infty} \frac{(-1)^m z^{r(2m+1)}}{(2m+1)!} \right| \\
&\geq |z^r| \left| \sum_{m=1}^{\infty} \frac{(-1)^m z^{2mr}}{(2m+1)!} \right| \\
&\geq |z^r||\tau_1|
\end{aligned}
$$

where $|\tau_1| \in (0,1]$ and $|\tau_1| \leq \left| \sum_{m=1}^{\infty} \frac{(-1)^m z^{2mr}}{(2m+1)!} \right|$.

Similarly,

$$|\sin(w^r)| \geq |w^r||\tau_2|, \text{ where } |\tau_2| \in (0,1] \text{ and } |\tau_2| \leq \left| \sum_{m=1}^{\infty} \frac{(-1)^m w^{2mr}}{(2m+1)!} \right|,$$

$$|\sin(v^r)| \geq |v^r||\tau_3|, \text{ where } |\tau_3| \in (0,1] \text{ and } |\tau_3| \leq \left| \sum_{m=1}^{\infty} \frac{(-1)^m v^{2mr}}{(2m+1)!} \right|,$$

$$|\sin(u^r)| \geq |u^r||\tau_4|, \text{ where } |\tau_4| \in (0,1] \text{ and } |\tau_4| \leq \left| \sum_{m=1}^{\infty} \frac{(-1)^m u^{2mr}}{(2m+1)!} \right|.$$

**Theorem 1.** *Let $T_c(z) = a\sin(z^r) + bz + c$ be a function. Consider $|z| \geq |c| > \max \left\{ \left( \frac{2 + |b|}{\gamma_1 |a||\tau_1|} \right)^{\frac{1}{r-1}}, \left( \frac{2 + |b|}{\beta_1 |a||\tau_2|} \right)^{\frac{1}{r-1}}, \left( \frac{2 + |b|}{|a|(\alpha_1 |\tau_3| - |\tau_2|)} \right)^{\frac{1}{r-1}}, \left( \frac{2 + |b|}{|a||\tau_4|} \right)^{\frac{1}{r-1}} \right\}$, where $0 < \alpha_1, \beta_1, \gamma_1 < 1$, and $c \in \mathbb{C}$. Define*

$$
\begin{aligned}
z_{n+1} &= T_c(u_n), \\
u_n &= (1 - \alpha_1)\, T_c(w_n) + \alpha_1 T_c(v_n), \\
v_n &= (1 - \beta_1)\, w_n + \beta_1 T_c(w_n), \\
w_n &= (1 - \gamma_1)\, z_n + \gamma_1 T_c(z_n),
\end{aligned}
\tag{2}
$$

*then* $|z_n| \to \infty$ *as* $n \to \infty$.

**Proof.** For $T_c(z) = a \sin(z^r) + bz + c$ and $n = 0$,

$$
\begin{aligned}
|w| &= |(1 - \gamma_1)\, z + \gamma_1 T_c(z)| \\
&= |(1 - \gamma_1)\, z + \gamma_1(a\sin(z^r) + bz + c)| \\
&\geq |\gamma_1(a\sin(z^r) + bz) + (1 - \gamma_1)z| - |\gamma_1 c| \\
&\geq \gamma_1|a||\sin(z^r)| - \gamma_1|b||z| - |z| + \gamma_1|z| - \gamma_1|c|.
\end{aligned}
$$

Now, by using $|z| \geq |c|$, we have

$$
\begin{aligned}
|w| &\geq \gamma_1|a||z^r||\tau_1| - \gamma_1|b||z| - |z| \\
&\geq \gamma_1|a||z^r||\tau_1| - |b||z| - |z| \\
&= |z|\big(\gamma_1|a||z^{r-1}||\tau_1| - |b| - 1\big).
\end{aligned}
$$

Since $|z| > \left(\frac{2+|b|}{\gamma_1|a||\tau_1|}\right)^{\frac{1}{r-1}}$ implies $\gamma_1|a||z^{r-1}||\tau_1| > 2 + |b|$, so

$$
\gamma_1|a||z^{r-1}||\tau_1| - |b| - 1 > 1.
$$

Therefore,

$$
|w| > |z|. \tag{3}
$$

Now, in the second step of (2), we have

$$
\begin{aligned}
|v| &= |(1 - \beta_1)\, w + \beta_1 T_c(w)| \\
&= |(1 - \beta_1)\, w + \beta_1(a\sin(w^r) + bw + c)| \\
&\geq |\beta_1(a\sin(w^r) + bw) + (1 - \beta_1)w| - |\beta_1 c| \\
&\geq |\beta_1(a\sin(w^r) + bw)| - (1 - \beta_1)|w| - |\beta_1 c| \\
&\geq \beta_1|a||w^r||\tau_2| - \beta_1|b||w| - |w| + \beta_1|w| - \beta_1|c|.
\end{aligned}
$$

By using (3), we obtain

$$
\begin{aligned}
|v| &\geq \beta_1|a||z^r||\tau_2| - \beta_1|b||z| - |z| \\
&\geq \beta_1|a||z^r||\tau_2| - |b||z| - |z| \\
&= |z|\big(\beta_1|a||z^{r-1}||\tau_2| - |b| - 1\big).
\end{aligned}
$$

Since $|z| > \left(\frac{2+|b|}{\beta_1|a||\tau_2|}\right)^{\frac{1}{r-1}}$ implies $\beta_1|a||z^{r-1}||\tau_2| > 2 + |b|$, so

$$
\beta_1|a||z^{r-1}||\tau_2| - |b| - 1 > 1.
$$

Hence,

$$
|v| > |z|. \tag{4}
$$

In the third step of (2), we have

$$
\begin{aligned}
|u| &= |(1-\alpha_1)\, T_c(w) + \alpha_1 T_c(v)| \\
&= |(1-\alpha_1)\,(asin(w^r) + bw + c) + \alpha_1(asin(v^r) + bv + c)| \\
&\geq \alpha_1|asin(v^r) + bv| - \alpha_1|c| - (1-\alpha_1)|asin(w^r) + bw| - (1-\alpha_1)|c| \\
&\geq \alpha_1|a||sin(v^r)| - \alpha_1|b||v| - \alpha_1|c| - (1-\alpha_1)|a||sin(w^r)| - (1-\alpha_1)|b||w| - (1-\alpha_1)|c| \\
&\geq \alpha_1|a||sin(v^r)| - \alpha_1|b||v| - \alpha_1|c| - |a||sin(w^r)| + \alpha_1|a||sin(w^r)| - |b||w| + \alpha_1|b||w| - |c| + \alpha_1|c|.
\end{aligned}
$$

By neglecting $\alpha_1|a||sin(w^r)|$ and using Equations (3) and (4) and $|z| \geq |c|$, we obtain

$$
\begin{aligned}
|u| &\geq \alpha_1|a||z^r||\tau_3| - \alpha_1|b||z| - \alpha_1|z| - |a||z^r||\tau_2| - |b||z| + \alpha_1|b||z| - |z| + \alpha_1|z| \\
&= \alpha_1|a||z^r||\tau_3| - |a||z^r||\tau_2| - |b||z| - |z| \\
&= |z|(\alpha_1|a||z^{r-1}||\tau_3| - |a||z^{r-1}||\tau_2| - |b| - 1).
\end{aligned}
$$

Since $|z| > \left(\frac{2+|b|}{|a|(\alpha_1|\tau_3| - |\tau_2|)}\right)^{\frac{1}{r-1}}$ implies $|z^{r-1}||a|(\alpha_1|\tau_3| - |\tau_2|) > 2 + |b|$, so

$$
\alpha_1|a||z^{r-1}||\tau_3| - |a||z^{r-1}||\tau_2| - |b| - 1 > 1.
$$

Thus,

$$
|u| > |z|.
$$

Now in the last step, for $z_{n+1} = T_c(u_n)$, we have

$$
\begin{aligned}
|z_1| &= |T_c(u)| \\
&= |asin(u^r) + bu + c| \\
&\geq |asin(u^r) + bu| - |c| \\
&\geq |a||u^r||\tau_4| - |b||u| - |z| \\
&\geq |a||z^r||\tau_4| - |b||z| - |z| \\
&\geq |z|(|a||z^{r-1}||\tau_4| - |b| - 1).
\end{aligned}
$$

When applying similar arguments repeatedly, we obtain

$$
\begin{aligned}
|z_2| &\geq |z|(|a||z^{r-1}||\tau_4| - |b| - 1)^2 \\
|z_3| &\geq |z|(|a||z^{r-1}||\tau_4| - |b| - 1)^3 \\
&\quad\vdots \\
|z_n| &\geq |z|(|a||z^{r-1}||\tau_4| - |b| - 1)^n.
\end{aligned}
$$

Since $|z| > \left(\frac{2+|b|}{|a||\tau_4|}\right)^{\frac{1}{r-1}}$ implies $|a||z|^{r-1}|\tau_4| > 2 + |b|$ and therefore $|a||z^{r-1}||\tau_4| - |b| - 1 > 1$, hence we have $|z_n| \to \infty$ as $n \to \infty$. □

**Corollary 1.** *Assume that* $|z_j| > \max\left\{|c|, \left(\frac{2+|b|}{\gamma_1|a||\tau_1|}\right)^{\frac{1}{r-1}}, \left(\frac{2+|b|}{\beta_1|a||\tau_2|}\right)^{\frac{1}{r-1}}, \left(\frac{2+|b|}{|a|(\alpha_1|\tau_3| - |\tau_2|)}\right)^{\frac{1}{r-1}}, \left(\frac{2+|b|}{|a||\tau_4|}\right)^{\frac{1}{r-1}}\right\}$. *Then,* $|z_{j+1}| \geq (1+\lambda_1)^k|z_j|$ *and* $|z_{j+1}| \to \infty$ *as* $k \to \infty$.

*3.2. Escape Criterion for $T_c(z) = ae^{z^r} + bz + c$*

For $r \geq 2$ and $c \in \mathbb{C}$, the series expansion for the exponential function is as follows:

$$
\begin{aligned}
|e^{z^r}| &= \left| 1 + z^r + \frac{z^{2r}}{2!} + \frac{z^{3r}}{3!} + \frac{z^{4r}}{4!} + \ldots \right| \\
&> \left| z^r + \frac{z^{2r}}{2!} + \frac{z^{3r}}{3!} + \frac{z^{4r}}{4!} + \ldots \right| \\
&= |z^r| \left| 1 + \frac{z^r}{2!} + \frac{z^{2r}}{3!} + \frac{z^{3r}}{4!} + \ldots \right| \\
&> |z^r||\tau_1|,
\end{aligned}
$$

where $|\tau_1| \in (0,1]$, so that $|\tau_1| < \left| 1 + \frac{z^r}{2!} + \frac{z^{2r}}{3!} + \frac{z^{3r}}{4!} + \ldots \right|$.

Similarly,

$$
|e^{w^r}| > |w^r||\tau_2|, \text{ where } |\tau_2| \in (0,1], \text{ so that } |\tau_2| < \left| 1 + \frac{w^r}{2!} + \frac{w^{2r}}{3!} + \frac{w^{3r}}{4!} + \ldots \right|,
$$

$$
|e^{v^r}| > |v^r||\tau_3|, \text{ where } |\tau_3| \in (0,1], \text{ so that } |\tau_3| < \left| 1 + \frac{v^r}{2!} + \frac{v^{2r}}{3!} + \frac{v^{3r}}{4!} + \ldots \right|,
$$

$$
|e^{u^r}| > |u^r||\tau_4|, \text{ where } |\tau_4| \in (0,1], \text{ so that } |\tau_4| < \left| 1 + \frac{u^r}{2!} + \frac{u^{2r}}{3!} + \frac{u^{3r}}{4!} + \ldots \right|.
$$

**Theorem 2.** *Let* $T_c(z) = ae^{z^r} + bz + c$ *be a function. Consider*

$$
|z| \geq |c| > \max \left\{ \left( \frac{2+|b|}{\gamma_1 |a||\tau_1|} \right)^{\frac{1}{r-1}}, \left( \frac{2+|b|}{\beta_1 |a||\tau_2|} \right)^{\frac{1}{r-1}}, \left( \frac{2+|b|}{|a|(\alpha_1|\tau_3|-|\tau_2|)} \right)^{\frac{1}{r-1}}, \left( \frac{2+|b|}{|a||\tau_4|} \right)^{\frac{1}{r-1}} \right\},
$$

*where* $0 < \alpha_1, \beta_1, \gamma_1 < 1$, *and* $c \in \mathbb{C}$. *Define*

$$
\begin{aligned}
z_{n+1} &= T_c(u_n), \\
u_n &= (1 - \alpha_1)\, T_c(w_n) + \alpha_1 T_c(v_n), \\
v_n &= (1 - \beta_1)\, w_n + \beta_1 T_c(w_n), \\
w_n &= (1 - \gamma_1)\, z_n + \gamma_1 T_c(z_n),
\end{aligned}
\tag{5}
$$

*then* $|z_n| \to \infty$ *as* $n \to \infty$.

**Proof.** For $T_c(z) = ae^{z^r} + bz + c$ and $n = 0$, we have

$$
\begin{aligned}
|w| &= |(1 - \gamma_1)\, z + \gamma_1 T_c(z)| \\
&= |(1 - \gamma_1)\, z + \gamma_1 (ae^{z^r} + bz + c)| \\
&\geq |\gamma_1(ae^{z^r} + bz) + (1 - \gamma_1)z| - |\gamma_1 c| \\
&\geq \gamma_1 |a||e^{z^r}| - \gamma_1 |b||z| - |z| + \gamma_1 |z| - \gamma_1 |c|.
\end{aligned}
$$

By using $|z| \geq |c|$, we have

$$
\begin{aligned}
|w| &\geq \gamma_1 |a||z^r||\tau_1| - \gamma_1 |b||z| - |z| \\
&\geq \gamma_1 |a||z^r||\tau_1| - |b||z| - |z| \\
&= |z| \left( \gamma_1 |a||z^{r-1}||\tau_1| - |b| - 1 \right).
\end{aligned}
$$

Since $|z| > \left( \frac{2+|b|}{\gamma_1 |a||\tau_1|} \right)^{\frac{1}{r-1}}$ implies $\gamma_1 |a||z^{r-1}||\tau_1| > 2 + |b|$, we obtain

$$
\gamma_1 |a||z^{r-1}||\tau_1| - |b| - 1 > 1.
$$

Therefore,

$$
|w| > |z|.
\tag{6}
$$

Similarly, in the second step of (5), we have

$$
\begin{aligned}
|v| &= |(1 - \beta_1)\, w + \beta_1 T_c(w)| \\
&= |(1 - \beta_1)\, w + \beta_1 (ae^{w^r} + bw + c)| \\
&\geq |\beta_1 (ae^{w^r} + bw) + (1 - \beta_1)w| - |\beta_1 c| \\
&\geq |\beta_1 (ae^{w^r} + bw)| - (1 - \beta_1)|w| - |\beta_1 c| \\
&\geq \beta_1 |a||w^r||\tau_2| - \beta_1 |b||w| - |w| + \beta_1 |w| - \beta_1 |z|.
\end{aligned}
$$

By using (6), we obtain

$$
\begin{aligned}
|v| &\geq \beta_1 |a||z^r||\tau_2| - \beta_1 |b||z| - |z| \\
&\geq \beta_1 |a||z^r||\tau_2| - |b||z| - |z| \\
&= |z|\left(\beta_1 |a||z^{r-1}||\tau_2| - |b| - 1\right).
\end{aligned}
$$

Since $|z| > \left(\frac{2+|b|}{\beta_1 |a||\tau_2|}\right)^{\frac{1}{r-1}}$ implies $\beta_1 |a||z^{r-1}||\tau_2| > 2 + |b|$ so

$$
\beta_1 |a||z^{r-1}||\tau_2| - |b| - 1 > 1.
$$

Therefore,

$$
|v| > |z|. \tag{7}
$$

Similarly, in the third step of (5), we obtain

$$
\begin{aligned}
|u| &= |(1 - \alpha_1)\, T_c(w) + \alpha_1 T_c(v)| \\
&= |(1 - \alpha_1)\, (ae^{w^r} + bw + c) + \alpha_1 (ae^{v^r} + bv + c)| \\
&\geq \alpha_1 |ae^{v^r} + bv| - \alpha_1 |c| - (1 - \alpha_1)|ae^{w^r} + bw| - (1 - \alpha_1)|c| \\
&\geq \alpha_1 |a||e^{v^r}| - \alpha_1 |b||v| - \alpha_1 |c| - (1 - \alpha_1)|a||ae^{w^r}| - (1 - \alpha_1)|b||w| - (1 - \alpha_1)|c| \\
&\geq \alpha_1 |a||e^{v^r}| - \alpha_1 |b||v| - \alpha_1 |c| - |a||e^{w^r}| + \alpha_1 |a||ae^{w^r}| - |b||w| + \alpha_1 |b||w| - |c| + \alpha_1 |c|.
\end{aligned}
$$

By neglecting $\alpha_1 |a||ae^{w^r}|$, and by using Equation (7) and $|z| \geq |c|$, we obtain

$$
\begin{aligned}
|u| &\geq \alpha_1 |a||z^r||\tau_3| - \alpha_1 |b||z| - \alpha_1 |z| - |a||z^r||\tau_2| - |b||z| + \alpha_1 |b||z| - |z| + \alpha_1 |z| \\
&= \alpha_1 |a||z^r||\tau_3| - |a||z^r||\tau_2| - |b||z| - |z| \\
&= |z|(\alpha_1 |a||z^{r-1}||\tau_3| - |a||z^{r-1}||\tau_2| - |b| - 1).
\end{aligned}
$$

Since $|z| > \left(\frac{2+|b|}{|a|(\alpha_1 |\tau_3| - |\tau_2|)}\right)^{\frac{1}{r-1}}$ implies $|z^{r-1}||a|(\alpha_1 |\tau_3| - |\tau_2|) > 2 + |b|$, we obtain

$$
\alpha_1 |a||z^{r-1}||\tau_3| - |a||z^{r-1}||\tau_2| - |b| - 1 > 1.
$$

As such, we have

$$
|u| > |z|. \tag{8}
$$

Now, in the final step for $z_{n+1} = T_c(u_n)$ and $n = 0$, we have

$$
\begin{aligned}
|z_1| &= |T_c(u)| \\
&= |ae^{u^r} + bu + c| \\
&\geq |ae^{u^r} + bu| - |c| \\
&\geq |a||u^r||\tau_4| - |b||u| - |z|.
\end{aligned}
$$

By using (8) and $|z| \geq |c|$, we obtain

$$
\begin{aligned}
|z_1| &\geq |a||z^r||\tau_4| - |b||z| - |z| \\
&\geq |z|(|a||z^{r-1}||\tau_4| - |b| - 1).
\end{aligned}
$$

By applying similar arguments repeatedly, we obtain

$$
\begin{aligned}
|z_2| &\geq |z|(|a||z^{r-1}||\tau_4| - |b| - 1)^2 \\
|z_3| &\geq |z|(|a||z^{r-1}||\tau_4| - |b| - 1)^3 \\
&\qquad . \\
&\qquad . \\
&\qquad . \\
|z_n| &\geq |z|(|a||z^{r-1}||\tau_4| - |b| - 1)^n.
\end{aligned}
$$

Since $|z| > \left(\frac{2+|b|}{|a||\tau_4|}\right)^{\frac{1}{r-1}}$ implies $|a||z|^{r-1}|\tau_4| > 2 + |b|$, we therefore have $|a||z^{r-1}||\tau_4| - |b| - 1 > 1$. Hence, $|z_n| \to \infty$ as $n \to \infty$. $\square$

**Corollary 2.** *Assume that* $|z_j| > \max\left\{|c|, \left(\frac{2+|b|}{\gamma_1|a||\tau_1|}\right)^{\frac{1}{r-1}}, \left(\frac{2+|b|}{\beta_1|a||\tau_2|}\right)^{\frac{1}{r-1}}, \left(\frac{2+|b|}{|a|(\alpha_1|\tau_3|-|\tau_2|)}\right)^{\frac{1}{r-1}}, \right.$
$\left.\left(\frac{2+|b|}{|a||\tau_4|}\right)^{\frac{1}{r-1}}\right\}$. *Then,* $|z_{j+1}| \geq (1+\lambda_1)^k |z_j|$ *and* $|z_{j+1}| \to \infty$ *are as* $k \to \infty$.

## 4. Algorithms

We present algorithms to create some fascinating fractals in this section. Using algorithms, we create source programs in MATHEMATICA 13.0 to generate the Julia and Mandelbrot sets (see Appendix A).

**Remark 1.** *Jet colormap, which used to be the default colormap in MATLAB, has been widely utilized in the literature for coloring fractals. In this work, we have made use of the Rainbow ColorFunction (a predefined ColorFunction in MATHEMATICA), which converts numerical values into a spectrum of colors—from red to violet to a variety of rainbow shades.*

**Remark 2.** *The Julia sets and Mandelbrot sets are both related to complex numbers and fractal geometry, but they are also distinct mathematical constructs with some key differences that can be clearly seen in Algorithms 1 and 2. In Algorithm 1, for the Mandelbrot set, we have a starting point/initial iteration of $z_0 = 0$. Meanwhile, in Algorithm 2, the Julia sets are typically studied by varying the initial values of $z_0$ for a given c to see which points remain bounded and which escape to infinity.*

---

**Algorithm 1** The Mandelbrot set.

---

**1. Setup:**
Take a complex number $c = u + iv$.
Initialize the variables $\alpha_1, \beta_1, \gamma_1, \tau_1, \tau_2, \tau_3, \tau_4, a, b$.
Set $z = c$.
**2. Iterate:**
$z_{n+1} = T_c(k_n)$;
$k_n = (1 - \alpha_1)\, T_c(m_n) + \alpha_1 T_c(l_n)$;
$l_n = (1 - \beta_1)\, m_n + \beta_1 T_c(m_n)$
$m_n = (1 - \gamma_1)\, z_n + \gamma_1 T_c(z_n); n \geq 0$,
where $T_c(z) = asin(z^r) + bz + c$ or $T_c(z) = ae^{z^r} + bz + c$, $r = 2, 3, 4, \ldots, 0 < \alpha_1, \beta_1, \gamma_1 < 1$, $0 < \tau_1, \tau_2, \tau_3, \tau_4 \leq 1$.
**3. Stop:**

$|z_n| >$Escape radius$= max\left\{|c|, \left(\frac{2+|b|}{\gamma_1|a||\tau_1|}\right)^{\frac{1}{r-1}}, \left(\frac{2+|b|}{\beta_1|a||\tau_2|}\right)^{\frac{1}{r-1}}, \left(\frac{2+|b|}{|a|(\alpha_1|\tau_3|-|\tau_2|)}\right)^{\frac{1}{r-1}}, \left(\frac{2+|b|}{|a||\tau_4|}\right)^{\frac{1}{r-1}}\right\}$
**4. Count:**
The number of iterations undertaken to escape.
**5. Color:**
Assign a color to each point based on the number of iterations needed to escape.

---

---

**Algorithm 2** The Julia set.

---

**1. Setup:**

Take a complex number $c = u + iv$.

Initialize the variables $\alpha_1, \beta_1, \gamma_1, \tau_1, \tau_2, \tau_3, \tau_4, a, b$.

Consider first iteration $z_0 = x + iy$.

**2. Iterate:**

$z_{n+1} = T_c(k_n)$;

$k_n = (1 - \alpha_1) T_c(m_n) + \alpha_1 T_c(l_n)$;

$l_n = (1 - \beta_1) m_n + \beta_1 T_c(m_n)$

$m_n = (1 - \gamma_1) z_n + \gamma_1 T_c(z_n); n \geq 0$,

where $T_c(z) = asin(z^r) + bz + c$ or $T_c(z) = ae^{z^r} + bz + c, r = 2, 3, 4, \ldots, 0 < \alpha_1, \beta_1, \gamma_1 < 1$,

$0 < \tau_1, \tau_2, \tau_3, \tau_4 \leq 1$.

**3. Stop:**

$|z_n| >$ Escape radius$= max\left\{ |c|, \left( \frac{2+|b|}{\gamma_1 |a||\tau_1|} \right)^{\frac{1}{r-1}}, \left( \frac{2+|b|}{\beta_1 |a||\tau_2|} \right)^{\frac{1}{r-1}}, \left( \frac{2+|b|}{|a|(\alpha_1|\tau_3|-|\tau_2|)} \right)^{\frac{1}{r-1}}, \left( \frac{2+|b|}{|a||\tau_4|} \right)^{\frac{1}{r-1}} \right\}$

**4. Count:**

Number of iterations undertaken to escape.

**5. Color:**

Based on the number of iterations needed to escape.

---

**Remark 3.** *One of the most evident symmetries in the Mandelbrot set is the symmetry with respect to the real axis. This means if there is a point $(a_1, b_1)$ where $a_1 = Re(c)$ and $b_1 = Im(c)$ belong to the Mandelbrot set, then so does the point $(a_1, -b_1)$. Similarly, if a point $(a_1, b_1)$, belongs to the Mandelbrot set, so is $(-a_1, b_1)$, which leads to vertical symmetry. In summary, the x-axis and y-axis symmetries of the Mandelbrot set arise from the fact that the Mandelbrot iteration depends on the magnitude of complex numbers rather than their individual components (both real and imaginary parts).*

*4.1. Mandelbrot Sets for $asin(z^r) + bz + c$*

In this subsection, we use the escape time approach to construct the Mandelbrot sets for the sine-associated function, which has a maximum of 10 iterations based on different input parameters. In Algorithm 1, the algorithm for creating Mandelbrot sets is described.

Case (i): For $T_c(z) = asin(z^r) + bz + c$, the parameter values vary, as given in Table 1:

**Table 1.** The parameters employed in Figure 1a–f.

| Figure | $r$ | $\tau_1$ | $\tau_2$ | $\tau_3$ | $\tau_4$ | $a$ | $b$ | $\alpha_1$ | $\beta_1$ | $\gamma_1$ |
|---|---|---|---|---|---|---|---|---|---|---|
| Figure 1a | 2 | 0.8 | 0.5 | 0.7 | 0.7 | 0.4 | 0.4 | 0.2 | 0.2 | 0.2 |
| Figure 1b | 2 | 0.08 | 0.05 | 0.07 | 0.07 | 0.4 | 0.4 | 0.8 | 0.8 | 0.8 |
| Figure 1c | 2 | 0.8 | 0.5 | 0.7 | 0.7 | 1 | 0 | 0.2 | 0.2 | 0.2 |
| Figure 1d | 3 | 0.08 | 0.05 | 0.07 | 0.07 | 1.14 | 0.9 | 0.9 | 0.9 | 0.9 |
| Figure 1e | 4 | 0.08 | 0.05 | 0.07 | 0.07 | 0.0014 | 0.0009 | 0.009 | 0.009 | 0.009 |
| Figure 1f | 6 | 0.08 | 0.05 | 0.07 | 0.07 | 0.0014 | 0.0009 | 0.009 | 0.009 | 0.009 |

In Figure 1a–c, quadratic Mandelbrot sets are presented, and we notice the change in shape by varying only the parameters $\tau_1, \tau_2, \tau_3$, and $\tau_4$, as well as by keeping the rest of them fixed. The resulting Mandelbrot sets are symmetrical along the $x$-axis, and they contain two main lobes. In Figure 1d, the Mandelbrot set is visualized for the polynomial $sinz^r + c$, and we keep all the parameters that are involved in the four-step feedback procedure equal. In addition, we noticed that the resulting Mandelbrot set was symmetrical to both the axes. In Figure 1e,f, for the polynomials of degree '$r$', we obtained $2r$ attractors at an angle of $\frac{m\pi}{r}$ (where '$m$' represents each attractor's position relative to the standard attractor). The parameters utilized in Figure 1a–f are listed in Table 1.

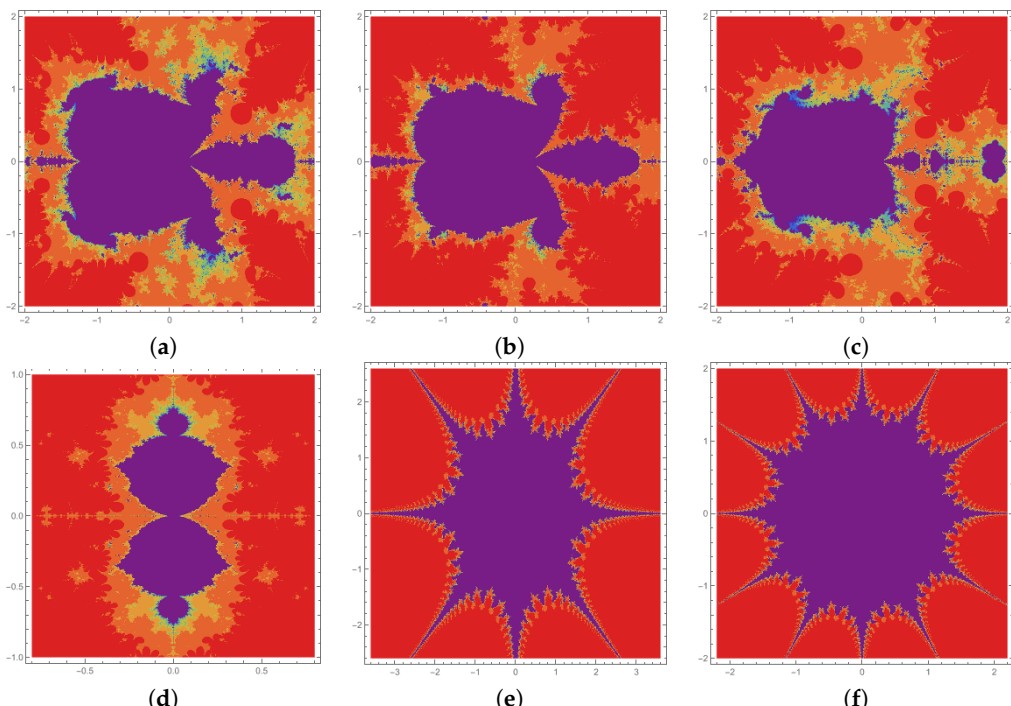

**Figure 1.** Effect of parameters on Mandelbrot set for sine function in PTO.

Case (ii): When *a* and *b* are complex numbers.

When complex values are assigned to the parameters '*a*' and '*b*', the quadratic Mandelbrot set, as shown in Figure 2a,b, resembles the classical Mandelbrot set, which has one bigger lobe and number of small lobes attached to it. Figure 2c,d represents two different versions of cubic Mandelbrot sets that have different shapes and sizes. In Figure 2e,f, we observed that, if only the parameter '*a*' is complex, then the number of bunches are $(r-1)$, and if both '*a*' and '*b*' are complex, then the number of bunches remains the same. The parameters utilized in Figure 2a–f are listed in Table 2.

**Table 2.** The parameters employed in Figure 2a–f.

| Figure | $r$ | $\tau_1$ | $\tau_2$ | $\tau_3$ | $\tau_4$ | $a$ | $b$ | $\alpha_1$ | $\beta_1$ | $\gamma_1$ |
|---|---|---|---|---|---|---|---|---|---|---|
| Figure 2a | 2 | 0.08 | 0.05 | 0.07 | 0.07 | 2.02 + 0.002i | 0.002i | 0.9 | 0.9 | 0.8 |
| Figure 2b | 2 | 0.8 | 0.5 | 0.7 | 0.7 | 1.0002i | 0.009i | 0.002 | 0.002 | 0.002 |
| Figure 2c | 3 | 0.08 | 0.05 | 0.07 | 0.07 | 0.014i | 0.009i | 0.09 | 0.09 | 0.09 |
| Figure 2d | 3 | 0.08 | 0.05 | 0.07 | 0.07 | 3.14 + 0.005i | 0.09 | 0.9 | 0.9 | 0.9 |
| Figure 2e | 6 | 0.08 | 0.05 | 0.07 | 0.07 | 1.14i | 0.9 | 0.9 | 0.9 | 0.9 |
| Figure 2f | 11 | 0.01 | 0.01 | 0.01 | 0.01 | −1.14i | −0.9i | 0.002 | 0.004 | 0.006 |

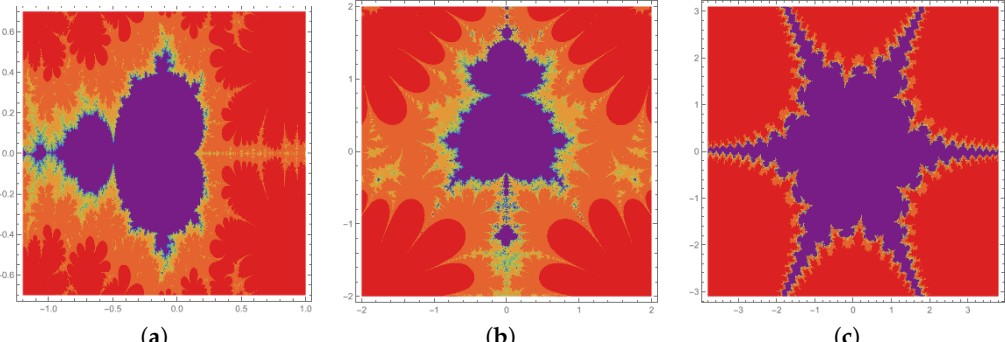

**Figure 2.** *Cont.*

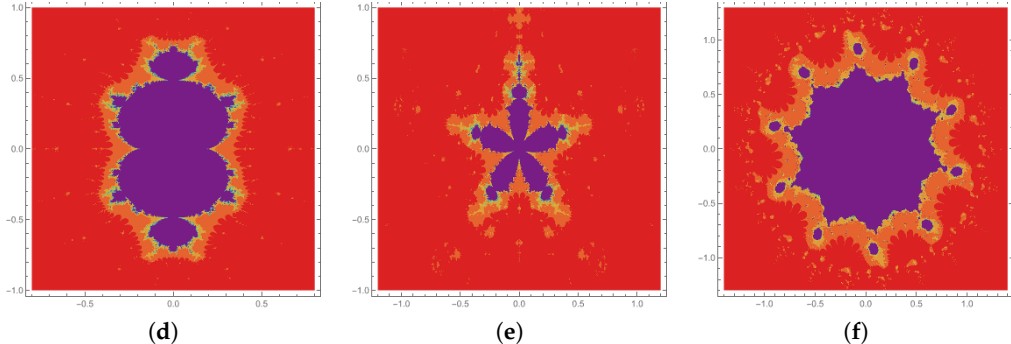

**Figure 2.** Effect of complex values of parameters *a* and *b* on Mandelbrot set for sine function.

### 4.2. Mandelbrot Sets for $ae^{z^r} + bz + c$

Here, the Mandelbrot sets for exponential functions are generated using the escape criteria. A variety of input parameters, as shown in Tables 2 and 3, were used for generating the fractals.

Case (i): When parameters *a* and *b* are real numbers.

**Table 3.** The parameters employed in Figure 3a–f.

| Figure | $r$ | $\tau_1$ | $\tau_2$ | $\tau_3$ | $\tau_4$ | $a$ | $b$ | $\alpha_1$ | $\beta_1$ | $\gamma_1$ |
|---|---|---|---|---|---|---|---|---|---|---|
| Figure 3a | 2 | 0.08 | 0.05 | 0.07 | 0.07 | 1.02 | 1.2 | 0.9 | 0.9 | 0.8 |
| Figure 3b | 3 | 0.08 | 0.05 | 0.07 | 0.07 | 1.02 | 1.2 | 0.2 | 0.2 | 0.2 |
| Figure 3c | 6 | 0.08 | 0.05 | 0.07 | 0.07 | 1.02 | 1.2 | 0.2 | 0.2 | 0.2 |
| Figure 3d | 2 | 0.000812 | 0.000575 | 0.000786 | 0.000775 | 1.02 | 1.2 | 0.9 | 0.9 | 0.8 |
| Figure 3e | 3 | 0.000812 | 0.000575 | 0.000786 | 0.000775 | 1.02 | 1.2 | 0.2 | 0.2 | 0.2 |
| Figure 3f | 6 | 0.000812 | 0.000575 | 0.000786 | 0.000775 | 1.02 | 1.2 | 0.2 | 0.2 | 0.2 |

In Figure 3a–f, we noticed a change in behavior in the Mandelbrot set corresponding to the exponential-associated function by varying the parameters involved in the Maclaurin's expansion of the exponential function. We observed that, by decreasing the values of these parameters, a noticeable change appeared in the color of Mandelbrot sets, while the shape remained unchanged. Also, by decreasing the value increases, the images' vibrancy and clarity similarly reduced. There are an '*r*' number of bunches, as well as the same number of Mandelbrot set junctions for a given value of '*r*'. The values listed in Table 3 correspond to the parameters used in Figure 3a–f.

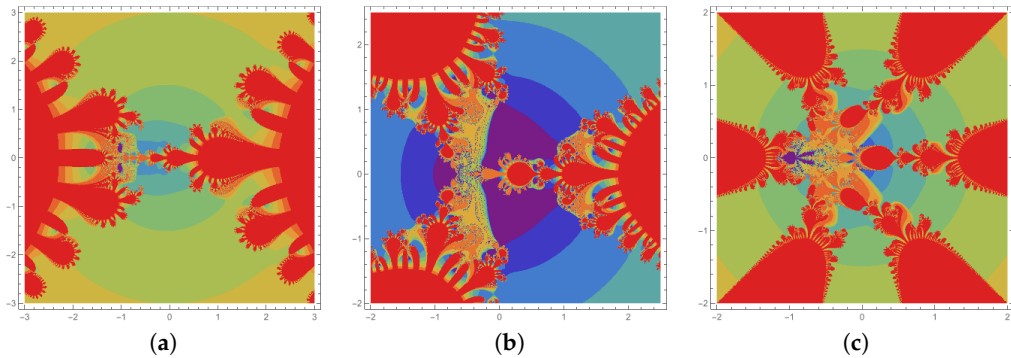

**Figure 3.** *Cont.*

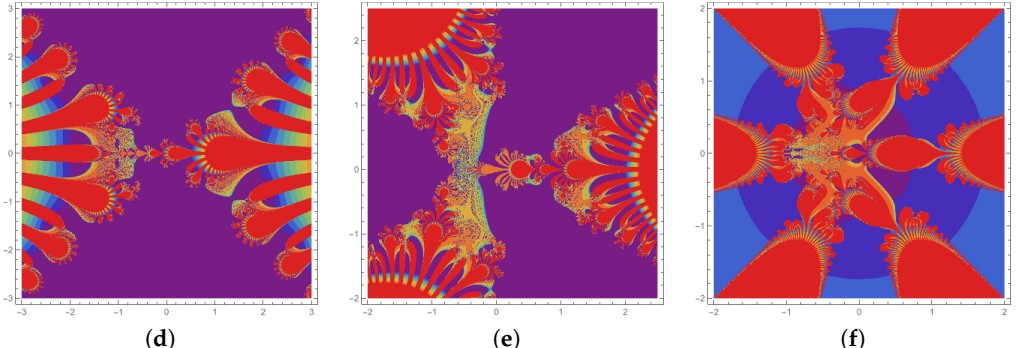

**Figure 3.** Effect of various parameters on Mandelbrot set for exponential function in PTO.

Case (ii): When $a$ and $b$ are complex numbers.

In Figure 4a–c, by keeping all the parameters fixed and choosing '$a$' and '$b$' as the complex numbers, we observe that, for a given value of '$r$', we obtain '$r$' copies of the pyramid-shaped Mandelbrot sets, which are joined end to end. The values listed in Table 4 correspond to the parameters used in Figure 4a–c.

**Table 4.** The parameters employed in Figure 4a–c.

| Figure | $r$ | $\tau_1$ | $\tau_2$ | $\tau_3$ | $\tau_4$ | $a$ | $b$ | $\alpha_1$ | $\beta_1$ | $\gamma_1$ |
|---|---|---|---|---|---|---|---|---|---|---|
| Figure 4a | 2 | 0.000814 | 0.000545 | 0.000721 | 0.000748 | 0.004i | 1.3 + 0.004i | 0.09 | 0.08 | 0.06 |
| Figure 4b | 3 | 0.000814 | 0.000545 | 0.000721 | 0.000748 | 0.004i | 1.3 + 0.004i | 0.09 | 0.08 | 0.06 |
| Figure 4c | 4 | 0.000814 | 0.000545 | 0.000721 | 0.000748 | 0.004i | 1.3 + 0.004i | 0.09 | 0.08 | 0.06 |

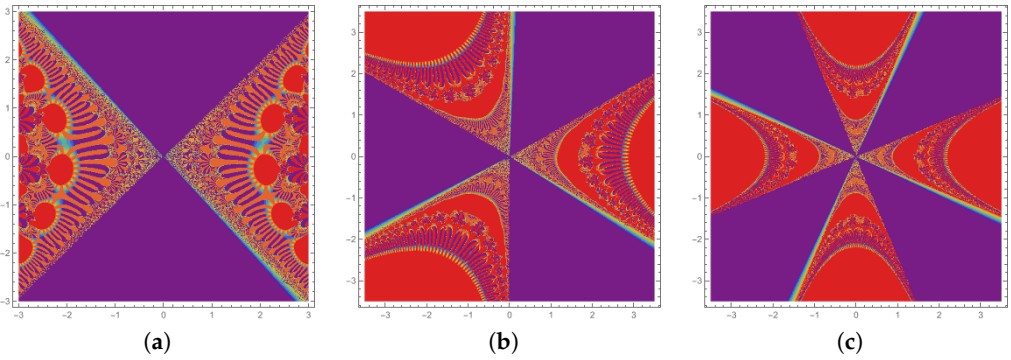

**Figure 4.** Effect of complex values of parameters $a$ and $b$ on Mandelbrot set for exponential function.

### 4.3. Julia Sets for $a\sin(z^r) + bz + c$

The change in behavior in the Julia sets for the sine-associated function is illustrated in this subsection. In Algorithm 2, the algorithm for creating the Julia sets is described.

In Figure 5a, we observed, for $r = 2$, there are four attractors at an angle of $\frac{m\pi}{2}$, where half of them are symmetric to the $x$-axis while remaining in the $y$-axis. As shown in Figure 5b,c, we slightly increased the respective values of the parameters '$a$' and '$b$', and we observed a tremendous change in the corresponding sets. As shown in Figure 5d–f, we noticed the effect of the varying parameters $\alpha_1, \beta_1,$ and $\gamma_1$ on the cubic Julia sets. We saw, by increasing the value of the said parameters, the number of bunches decrease, and, for $\alpha_1 = \beta_1 = \gamma_1 = 0.9$, it took the shape of a six-armed starfish. As shown in Figure 5g, the quadratic Julia set resembled a butterfly, whereas alluring graphics were obtained—again, as shown in Figure 5h,i—for $r = 4$ and $r = 8$, respectively. The values listed in Table 5 correspond to the parameters used in Figure 5a–i.

**Table 5.** The parameters employed in Figure 5a–i.

| Figure | $r$ | $\tau_1$ | $\tau_2$ | $\tau_3$ | $\tau_4$ | $a$ | $b$ | $c$ | $\alpha_1$ | $\beta_1$ | $\gamma_1$ |
|---|---|---|---|---|---|---|---|---|---|---|---|
| Figure 5a | 2 | 0.06 | 0.07 | 0.08 | 0.09 | 1 | 0 | $-0.007i$ | 0.01 | 0.02 | 0.03 |
| Figure 5b | 2 | 0.06 | 0.07 | 0.08 | 0.09 | 1 | 0.5 | $-0.007i$ | 0.01 | 0.02 | 0.03 |
| Figure 5c | 2 | 0.06 | 0.07 | 0.08 | 0.09 | 1.7 | 0 | $-0.007i$ | 0.01 | 0.02 | 0.03 |
| Figure 5d | 3 | 0.6 | 0.7 | 0.8 | 0.9 | 0.8 | 0.02 | $0.0007 - 0.0007i$ | 0.03 | 0.03 | 0.03 |
| Figure 5e | 3 | 0.6 | 0.7 | 0.8 | 0.9 | 0.8 | 0.02 | $0.0007 - 0.0007i$ | 0.5 | 0.5 | 0.5 |
| Figure 5f | 3 | 0.6 | 0.7 | 0.8 | 0.9 | 0.8 | 0.02 | $0.0007 - 0.0007i$ | 0.9 | 0.9 | 0.9 |
| Figure 5g | 4 | 0.6 | 0.7 | 0.8 | 0.9 | 0.2 | 1.2 | $0.0008888$ | 0.07 | 0.05 | 0.08 |
| Figure 5h | 4 | 0.6 | 0.7 | 0.8 | 0.9 | 2.2 | 1.2 | $-0.00088 - 0.00088i$ | 0.07 | 0.05 | 0.08 |
| Figure 5i | 8 | 0.6 | 0.7 | 0.8 | 0.9 | 1.2 | 1.2 | $-0.0008i$ | 0.07 | 0.05 | 0.08 |

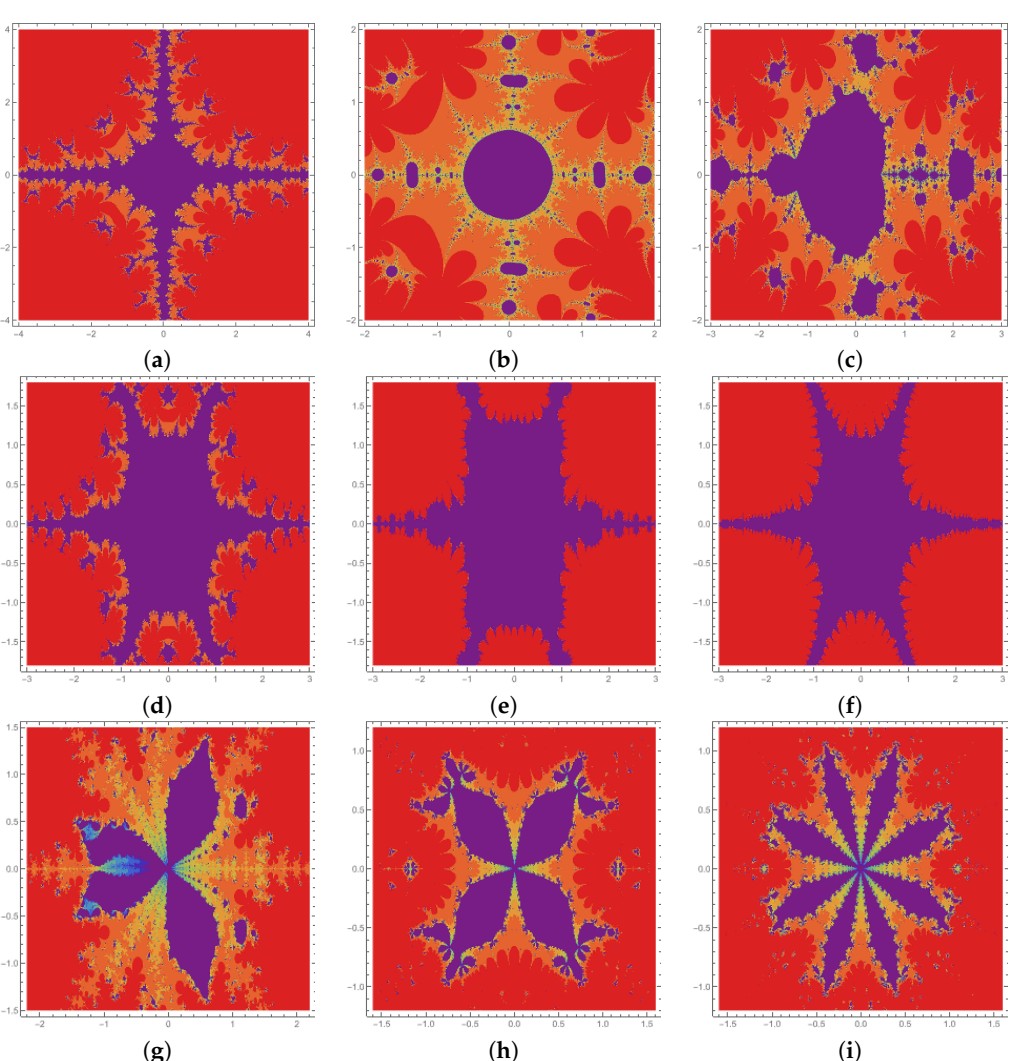

**Figure 5.** Effect of parameters on Julia set for sine function in PTO.

### 4.4. Julia Sets for $ae^{z^r} + bz + c$

The Julia sets corresponding to the transcendental exponential function for different input values are illustrated in this subsection.

Here, for $r = 2$, exponential function had two bunches, and the distance between them decreased as we increased the values of $\alpha_1, \beta_1,$ and $\gamma_1$, as seen in Figure 6a,b. Figure 6c shows that the Julia set became more amazing and vivid as we reduced the values of the $\tau_1, \tau_2, \tau_3,$ and $\tau_4$ parameters. In Figure 6d–f, the number of bunches increased by increasing the value of $r$. The values listed in Table 6 correspond to the parameters used in Figure 6a–f.

**Table 6.** The parameters employed in Figure 6a–f.

| Figure | $r$ | $\tau_1$ | $\tau_2$ | $\tau_3$ | $\tau_4$ | $a$ | $b$ | $c$ | $\alpha_1$ | $\beta_1$ | $\gamma_1$ |
|---|---|---|---|---|---|---|---|---|---|---|---|
| Figure 6a | 2 | 0.5 | 0.7 | 0.4 | 0.7 | 1 | 0 | $-0.5i$ | 0.5 | 0.7 | 0.7 |
| Figure 6b | 2 | 0.5 | 0.7 | 0.4 | 0.7 | 1 | 0 | $-0.5i$ | 0.9 | 0.9 | 0.9 |
| Figure 6c | 2 | 0.000580 | 0.000745 | 0.000456 | 0.000714 | 0.05 | 1.2 | $0.45 - 0.08i$ | 0.5 | 0.7 | 0.9 |
| Figure 6d | 3 | 0.7 | 0.9 | 0.3 | 0.5 | $-1$ | 0.001 | $0.4i$ | 0.4 | 0.4 | 0.4 |
| Figure 6e | 4 | 0.7 | 0.9 | 0.3 | 0.5 | $-1$ | 0.0001 | 2 | 0.0001 | 0.0001 | 0.0001 |
| Figure 6f | 6 | 0.07 | 0.09 | 0.03 | 0.05 | $-1$ | 0.0001 | 2 | 0.09 | 0.09 | 0.09 |

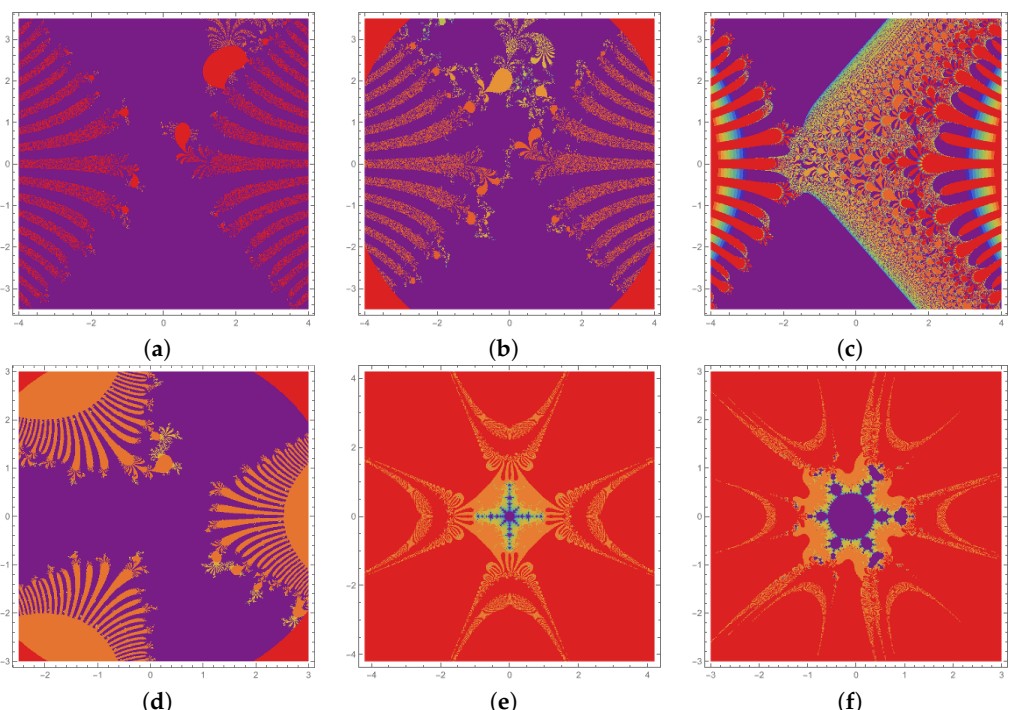

**Figure 6.** Effect of parameters on Julia set for exponential function in PTO.

## 5. Conclusions

In this study, by using the Picard–Thakur iteration, escape conditions were provided by examining the complex sine $T_c(z) = a\sin(z^r) + bz + c$ and exponential $T_c(z) = ae^{z^r} + bz + c$ functions. These findings were used in algorithms to depict the Julia and Mandelbrot sets. We used MATHEMATICA 13.0 to generate the appealing fractals and to explore the various regions of the complex plane, as well as to study the intricate patterns that form within the Julia set by altering the parameter $c$. It was observed that the parameters $\tau_1$, $\tau_2$, $\tau_3$, and $\tau_4$ had a significant impact on the color variation of the Julia sets for exponential functions. Also, for the complex values of the variation parameters, the sine function's quadratic Mandelbrot sets took the structure of regular Mandelbrot sets; meanwhile, in the case of the exponential functions, the complex parameters added more complexity to the Mandelbrot sets. In the future, we will try to generate fractals for complex cosine functions, as well as functions of the type $z^r + e^{c^t}$, by introducing metrics like generation time and ANI while using a fixed-point iterative process.

**Author Contributions:** Conceptualization, A.B., A.P. and M.S.; methodology, A.B.; formal analysis, A.B.; investigation, A.P.; writing—original draft preparation, A.B.; writing—review and editing, A.B. and M.S.; visualization, A.B.; supervision, A.P. and M.S. All authors have read and agreed to the published version of the manuscript.

**Funding:** This research received no external funding.

**Data Availability Statement:** Not applicable.

**Conflicts of Interest:** The authors declare no conflict of interest.

## Appendix A

*Appendix A.1. Source Program to Generate the Julia Sets*

$iter[x, y, lim] = Block[\{c, z, ct, a, b, \alpha, \beta, \gamma, \tau_1, \tau_2, \tau_3, \tau_4\},$
$c = m + nI;$
$z = x + yI;$
$\tau_1 = 0.8;$
$\tau_2 = 0.5;$
$\tau_3 = 0.7;$
$\tau_4 = 0.7;$
$\alpha_1 = 0.9;$
$\beta_1 = 0.8;$
$\gamma_1 = 0.75;$
$a = 0.4;$
$b = 0.4;$
$ct = 0;$
$m = 0.01;$
$n = 0.01;$
$While[(Abs[z] < max\{|c|, \left(\frac{2+|b|}{\gamma_1|a||\tau_1|}\right)^{\frac{1}{r-1}}, \left(\frac{2+|b|}{\beta_1|a||\tau_2|}\right)^{\frac{1}{r-1}}, \left(\frac{2+|b|}{|a|(\alpha_1|\tau_3|-|\tau_2|)}\right)^{\frac{1}{r-1}}, \left(\frac{2+|b|}{|a||\tau_4|}\right)^{\frac{1}{r-1}}\}) \&\&$
$(ct <= lim), ++ct;$
$g = a * Sin[z^r] + b * z + c;$
$w = (1 - \gamma) * z + \gamma * g;$
$f = a * Sin[w^r] + b * w + c;$
$v = (1 - \beta) * w + \beta * f;$
$d = a * Sin[v^r] + b * v + c;$
$u = (1 - \alpha) * f + \alpha * d;$
$z = a * Sin[u^r] + b * u + c;];$
$Return[ct];$
$]$
$DensityPlot [-iter [x, y, no. of iterations], \{x, xmin, xmax\}, \{y, ymin, ymax\}, PlotPoints \rightarrow$
$200, Mesh \rightarrow False,$
$ColorFunction \rightarrow "Rainbow", PlotLegends \rightarrow Automatic]$

*Appendix A.2. Source Program to Generate Mandelbrot Sets*

$iter[x, y, lim] = Block[\{c, z, ct, a, b, \alpha, \beta, \gamma, \tau_1, \tau_2, \tau_3, \tau_4\},$
$c = x + yI;$
$z = c;$
$\tau_1 = 0.8;$
$\tau_2 = 0.5;$
$\tau_3 = 0.7;$
$\tau_4 = 0.7;$
$\alpha_1 = 0.9;$
$\beta_1 = 0.8;$
$\gamma_1 = 0.75;$
$a = 0.4;$
$b = 0.4;$
$ct = 0;$
$While[(Abs[z] < max\{|c|, \left(\frac{2+|b|}{\gamma_1|a||\tau_1|}\right)^{\frac{1}{r-1}}, \left(\frac{2+|b|}{\beta_1|a||\tau_2|}\right)^{\frac{1}{r-1}}, \left(\frac{2+|b|}{|a|(\alpha_1|\tau_3|-|\tau_2|)}\right)^{\frac{1}{r-1}}, \left(\frac{2+|b|}{|a||\tau_4|}\right)^{\frac{1}{r-1}}\}) \&\&$
$(ct <= lim), ++ct;$

$$g = a * Sin[z^r] + b * z + c;$$
$$w = (1 - \gamma) * z + \gamma * g;$$
$$f = a * Sin[w^r] + b * w + c;$$
$$v = (1 - \beta) * w + \beta * f;$$
$$d = a * Sin[v^r] + b * v + c;$$
$$u = (1 - \alpha) * f + \alpha * d;$$
$$z = a * Sin[u^r] + b * u + c;];$$
$$Return[ct];$$
]

$DensityPlot[-iter[x, y, no.of iterations], \{x, xmin, xmax\}, \{y, ymin, ymax\}, PlotPoints \rightarrow 200, Mesh \rightarrow False,$
$ColorFunction \rightarrow "Rainbow", PlotLegends \rightarrow Automatic]$

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
