# Peer review of "Mandelbrot and Julia Sets of Transcendental Functions Using Picard–Thakur Iteration"

_fractalfract, doi:10.3390/fractalfract7100768_

Round 1

Reviewer 1 Report

The paper concerns the issue of constructing fractals using the Picard-Thakur hybrid iterative scheme. The complex-valued functions used are of the transcendental type. Two theorems are proposed and proved. Two algorithms and several examples are also provided. At the end of the manuscript, references cite previous papers appropriately. In general, it is an interesting manuscript.

Below are issues that I would like to see the authors address before the paper is published:

1- Although fractals are beautiful, their meaning needs further discussion. Please provide some clarification regarding the importance of the new fractals presented in the paper. 

2- The paper deals with alterations to the structures of the fractals. In this connection some interesting alterations have been made, notably including memory-based models (for instance, see the reference 10.25088/ComplexSystems.21.4.269). I feel that some discussion of how the results from the manuscript fit in with the concept of the “embedding” of memory in complex maps is warranted here.

3- The theorems give the conditions under which the stable regions exist, but no rationale whatsoever is provided regarding the symmetry of the sets. In this respect, where possible, detailed calculations should be incorporated in the manuscript.

4- Nowadays, recent papers also explore various metrics that can be used to assess dependencies between parameters. Two metrics are commonly measured (for instance, see the reference 10.1016/j.matcom.2023.02.012): generation time and average number of performed iteration. I think it would be very interesting if the authors could make calculations of such metrics (in a manner that is tied to the illustrative examples provided).

One technical issue to be dealt with is the repetition of identical equations which increase the length of proofs. In particular, I refer to two sets of equations: set 1 - Eqs. (2), (3), (4), and (5), and set 2 - Eqs. (10), (11), (12), and (13). For each set, only one equation is needed.

I recommend a review of the manuscript for language quality.

Author Response

Thanks for reviewing manuscript. Please see responses by authors in attached file.

Reviewer 2 Report

The concepts  of escape  and escape criteria should be defined explicitly or better characterized with wording sentences. The  statements of some of the given, and  mathematically proved, technical  results should be improved by rewritting some of the sentences  more formally, by writing some of the current separate conditions in unique combined constraints when possible( for instance , by using max(. , . , . ,.) , etc..

Explain the concept of escape with words in the abstract and introduction.

Are the given conditions for escape  in the various results necessary , sufficient or necessary and sufficient?. It seems that they are just sufficient since that taus/i are characterized  just through  bounds but not explicitly. So, the conditions for escape are not fully covered.

Eqn. next to (17): Add a left hand-side term abs(z1) which is now in line with a wording written text . Revise similar point in other places like, for instance, at the end of page 4.

Statement of Theorem 3.1 : It can be better to rewrite the conditions together ( for instance, rewrite second right-hand-side as a maximum of four expressions and a unique left-hand-sige being non less than abs(c)).

Final expression on page 5, add left-hand-side to the inequality within the formula which is now in the  previous text line.

Theorem 3.2: there is absolute  value of c missed in the second line of the statement.

Which are the differences between Algorithm  1 and Algorithm 2?. Please comment on them n a remark.

Author Response

(The authors gave the same response as above.)

Round 2

Reviewer 1 Report

I have read the revised version of the manuscript. To be clear, I don't feel that the authors have done enough to persuade the reader that the manuscript is worthy of publication in MDPI after a revision was made. In my humble opinion, the impact of the research is actually quite modest. More work would be needed to plumb the potential of the ideas presented.

I recommend a review of the manuscript for language quality.